# Estrogen Receptor Beta: The Promising Biomarker and Potential Target in Metastases

**DOI:** 10.3390/ijms22041656

**Published:** 2021-02-06

**Authors:** Ana Božović, Vesna Mandušić, Lidija Todorović, Milena Krajnović

**Affiliations:** Laboratory for Radiobiology and Molecular Genetics, Vinča Institute of Nuclear Sciences, National Institute of the Republic of Serbia, University of Belgrade, P.O. Box 522, 11001 Belgrade, Serbia; vvranic@vin.bg.ac.rs (V.M.); lidijat@vin.bg.ac.rs (L.T.); mdragic@vin.bg.ac.rs (M.K.)

**Keywords:** Estrogen Receptor Beta, endocrine-related cancers, carcinogenesis, biomarker, metastases, tumor-suppressor, isoforms, promoters, methylation

## Abstract

The discovery of the Estrogen Receptor Beta (ERβ) in 1996 opened new perspectives in the diagnostics and therapy of different types of cancer. Here, we present a review of the present research knowledge about its role in endocrine-related cancers: breast, prostate, and thyroid, and colorectal cancers. We also discuss the reasons for the controversy of its role in carcinogenesis and why it is still not in use as a biomarker in clinical practice. Given that the diagnostics and therapy would benefit from the introduction of new biomarkers, we suggest ways to overcome the contradictions in elucidating the role of ERβ.

## 1. Introduction

After more than two decades since the discovery of the second receptor for estrogen, Estrogen Receptor Beta (Erβ) [1,2], its role in etiology of different malignancies is still controversial. The role of ERβ has been implicated in many cancer types [3,4]. Despite the intensive investigation and numerous studies published to date, most of the data on ERβ expression are still controversial. Similar to the other steroid receptors, ERβ is expressed as a pool of isoforms with different C-termini, together with several splice variants generated by the precise deletion of the internal exons, as reviewed in [5,6]. In earlier studies, antibodies targeted to the N-terminal region were used, and their findings mainly reflected the total ERβ transcript pool. Even upon the discovery of ERβ variants and isoforms, the majority of the studies were done with inadequately validated antibodies. Despite some researchers’ efforts to evaluate the specificity and sensitivity of commercially available antibodies [7,8], the controversial findings of the ERβ expression level in disease progression, prediction, prognosis, and therapy, still prevail.

ERβ, along with ERα, is a member of the superfamily of nuclear receptors. These receptors are coded with highly homologous genes. Genes for ERα (*ESR1*) and ERβ (*ESR2*) are located on different chromosomes, chromosome 6 and chromosome 14, coding for proteins that have 595 and 530 amino acids, respectively. Genes for ERs contain eight exons [9,10] and are alternatively transcribed into protein isoforms with different C-endings. The structure of the ERβ gene and the corresponding protein is shown in Figure 1. Besides the wild-type isoform ERβ1, there are four more isoforms of ERβ: ERβ2, ERβ3, ERβ4, and ERβ5. Exons 1 to 7 are the same, but exon 8 is specific to every isoform [11,12]. The isoforms alternatively splice exon 8 consequently forming proteins of a lower molecular weight than the wild-type protein. In the breast tissue, ERβ1, ERβ2, and ERβ5, are dominant, and it seems that they have different biological roles. ERβ3 is expressed in the prostate, while ERβ4 expression was observed in the breast [13,14]. There is a difference in the expression of isoforms in normal and cancer tissue. ERβ1 forms functional homodimers and heterodimers with the other isoforms. By forming heterodimers with ERα, it inhibits the ERα signaling pathway [12,15]. It is assumed that ERβ2 and ERβ5 do not act independently but form heterodimers with ERβ1 and ERα, therefore influencing their activity [16]. Obviously, it is important to have in mind the role of all ERβ isoforms in carcinogenesis. The main wt ERβ isoform and the others are depicted in Figure 2.

Despite their location on different chromosomes, estrogen receptors have highly homologous protein domains. Like the other members of the nuclear receptor superfamily, the ERβ protein has six functional domains, from the N-terminal A/B domain to the C-terminal F domain, with different degrees of sequence conservation [18]. A poorly conserved A/B domain codes for a protein part with a transactivation function (AF-1). The C domain is highly conserved and contains a DNA binding domain (DBD). A conserved E-domain codes for the ligand-binding domain and transactivation function AF-2 [19]. The A D domain links the DBD and LBD domains. The F domain is in the C-terminal region, continues to the LBD, and is not conserved [20]. Because exons 1–7 are the same, all ERβ isoforms have the same AF1, DBD, D, and LBD domains, while the last AF2 domain on the C-terminus is specific to every isoform [21]. The difference in the AF2 functional domains of the isoforms leads to a change in protein activity or a total lack of it [2]. Corresponding ligand binding causes a conformational change, which leads to receptor dimerization and binding to a specific sequence on the DNA called “Estrogen Response Elements” (EREs). An activated receptor/DNA complex interacts with the other nuclear cofactors, which lead to DNA transcription downstream from EREs and the activation of estrogen-regulated genes [22]. DNA-binding domains are 96% homologous in estrogen receptors and bind identically to the majority of EREs [23]. However, transcription regulation depends on the estrogen receptor subtype because such receptors bind to transcription factors differently [9,24]. The LBD domains of ERα and ERβ are 54% homologous and bind estrogens and selective estrogen receptor modulators (SERMs) [25]. Activation domains AF1 and AF2 synergistically interact with different coactivators to bind to the DNA/ER complex [23]. Unlike ERα, where coactivators simultaneously bind to AF1 and AF2 to reach full activation, ERβ has a lower activity of AF1 but a completely functional AF2 [26]. The functional domains of the ERβ protein are shown in Figure 1.

## 2. The Molecular Pathways of Estrogen Receptor Alpha and Beta in Cancer

Decades ago, it became clear that estrogen has a carcinogenic effect. In the meantime, it has been discovered that it has a dual role in the proliferation and the cell cycle [27,28,29]. The discovery of ERβ means that the dual role of estrogen is possibly mediated through two receptors, ERα and ERβ, and probably through the complex profile of its isoforms [30,31,32,33].

The classical, direct genomic action of estrogen starts upon ligand binding to the cytosolic receptor, followed by nuclear translocation, dimerization, and binding to ERE sites in target genes. Other estrogen-dependent pathway signals originate from a pool of membrane ERs, and this pathway is triggered by ligand binding and mediated through signaling cascades (Akt, PKA, and ERK1/2) and usually activated STAT, CREB, NF-kB, and Jun transcription factors. Although both receptors, ERα and ERβ, share a similar domain structure and can bind estrogen and initiate transcription from ERE sites, in the case of ERβ, the activation function of the AF-1 domain is weaker than that of ERα, while the AF-2 function of ERβ is similar to ERα. In addition to the ligand activation, by tethering to the other transcription factors, such as AP1, SP1, or NF-κB, ERβ activates the expression of a unique sets of genes (ligand-independent action in the absence of estrogen), and this pathway is usually related to growth factor signaling via activated kinases. According to the study of Acconcia and coworkers, it seems that the main difference in the specific activation of survival pathways mediated by ERα and apoptotic pathways mediated by ERβ is through its non-genomic, membrane-initiated pathways. In a study on HeLa, HepG2, and DLD1 cell lines containing transfected or endogenous ERα and ERβ, the authors showed that, after estrogen stimulation, the ERα rapidly activates multiple signal transduction pathways (ERK/MAPK and PI3K/AKT) related to the cell cycle progression and prevention of the apoptotic cascade, while ERβ induces the rapid and persistent phosphorylation of p38/MAPK, which is involved in caspase-3 activation and cleavage of poly(ADP-ribose)polymerase, driving cells into the apoptotic cycle. They also showed that ERβ did not activate any of the ERα-mediated pathways of cell growth signaling [30]. Another important study was reported by Helguero and coworkers on HC11 mammary epithelial cells that express both receptors, treated with ERα- and ERβ-specific new developed agonists [31]. They showed that, after exposure to the ERα-selective agonist, 4,4′,4″-(4-propyl-(1H)-pyrazole-1,3,5-triyl)trisphenol (PPT), there was a 50% increase in cell number, but 2,3-bis(4-hydroxy-phenyl)-propionitrile (DPN)-Erβ-selective agonist decreased the cell number by 20–30%. After exposure to E2, the cell number was unchanged. They also showed that E2 and PPT treatment increased cyclin D1 expression, while DPN did not. The expression of proliferating cell nuclear antigen (PCNA) expression also decreased following DPN treatment, as well as the mitotic index and Ki67 expression. Their study demonstrates that estrogen signaling via ERα favors entry into the cell cycle, but ERβ transiently inhibits it, and the loss of ERβ expression favors cell transformation.

In line with in vitro studies are findings from association studies on clinical samples of prostate, breast, and thyroid carcinomas. The results of these studies showed that the ERα/ERβ ratio changes, with ERα being upregulated and ERβ being downregulated [34,35,36,37,38]. It has become clear that the antiproliferative activity of ERβ qualifies this receptor as an important therapeutic target, in cancers that involve estrogen signaling (breast, prostate, and colon) and in those that are gender-related (stomach, chronic lymphocytic leukemia, and thyroid) as reviewed in [16].

## 3. Epigenetic Regulation of ERβ

Epigenetic regulation is crucial for many biological processes: chromosome X inactivation, genomic imprinting, RNA interference, and reprogramming of the genome during differentiation and development [39]. The disruption of these processes manifests through aberrant gene methylation and/or histone acetylation, which leads to alterations in gene expression. The loss of cell adhesion proteins and the excessive stimulation of signaling pathways of estrogen receptors lead to abnormal growth and differentiation of tissue. Migration of cells increases; many intracellular pathways of apoptosis, DNA reparation, and detoxification are activated [40]. The disrupted epigenetic processes cause many diseases, including cancer.

One of the most important epigenetic mechanisms that lead to cancer initiation and progression is the methylation of CpG islands. CpG islands are located in the promoter regions and sometimes in the first exons of genes, spanning from 500 to several thousands of nucleotides. The frequency of CpG dinucleotides in these islands is higher than in other gene locations [41,42]. Non-methylated CpG islands are usually situated in the promoter regions of housekeeping genes, essential for the general functions of the cells, and in some tissue-specific genes [43]. Excessive activity of DNA methyltransferases (DNMT) leads to CpG islands hypermethylation in cancer DNMT catalyzes the transfer of the methyl group from S adenosyl L-methionine (SAM) to 5′-cytosines of CpGs [40]. The most important human DNMTs are DNMT1, DNMT3a, and DNMT3b. The most abundant is DNMT1, which sustains the methylation pattern. The other two mediate in de novo methylation [44]. The consequence of hypermethylation is the reversible inactivation of tumor suppressor genes, which is a heritable change that passes to the next generation of cells through mitosis [40]. In cancer cells, the global hypomethylation of internal exons and introns of genomic DNA and the hypermethylation of nonmethylated regions of gene promoters lead to methylation imbalance [45]. Methylation of CpG islands in promoter regions leads to long-term gene silencing and chromatin unavailability for transcription [46]. In hormone-dependent cancers, the expression of estrogen receptors is associated with clinical outcomes [4]. For example, in breast cancer, the methylation of specific gene clusters leads to the expression or absence of ERs and PR, causing metastases and relapse of the disease [47]. ERα is expressed in 75% of breast cancers, and ERα-positive patients better respond to therapy and therefore have a better prognosis [48]. ERα-negative patients have a worse prognosis. One of the assumed mechanisms of a lack of ERα is hypermethylation of the ERα gene promoter. There is no methylation in normal breast tissue in these regions. Hypermethylation of CpG islands occurs in all tumor stadiums with a higher frequency in the transition phase from the ductal carcinoma in situ to metastasis [49]. The expression of ERβ also changes during the progression of breast cancer. In the initial phases of the disease, the receptor level decreases, to be lost in the advanced stages of the disease [50,51,52]. On the other hand, there is a high level of ERβ in most prostate metastases, in the bones, and in lymph nodes [53]. The role of ERβ in metastases is unclear. It could be assumed that local factors lead to the re-expression of ERβ in metastases [54].

Li and coworkers cloned and characterized the promoter region of human ERβ in 2000 [55]. Soon afterwards, Hirata and al. described the two isoforms originating from the first two non-translational exons of the ERβ gene, exon OK and ON. These exons splice to exon 1 and form two isoforms, OK-1 and ON-1. Exon OK is located 50 Kb upstream of exon ON [56]. The scientists also described whole-length transcripts that have neither the exon ON nor the exon OK [57]. In 2001, Smith and coworkers discovered the third promoter region, which transcribes to the functional transcript [58]. The distribution of the OK and ON isoforms in the tissues is different, and it changes in cancer cells [56,59]. 

ERβ is a dominant form of ER in the prostate and the increased promoter methylation is primarily discovered in prostate cancer [54]. Nojima et al. showed that methylation in the 5′-untranslated region correlated inversely with ERβ expression in the prostate. They analyzed 19 CpG places between 376 and 117 bp upstream from ATG, in exon ON, in human prostate carcinoma, and in benign prostate hyperplasia (BPH). All 19 CpG places were methylated, and associated with the loss of ERβ expression. In BPH, CpGs were not methylated, and there was ERβ expression. Treatment of prostate cancer cells with 5-aza-2-deoxicytidine (5-AZAC) led to the reexpression of ERβ, which proves that the methylation is a reversible process [60]. Zhu and al. included the promoter region upstream of exon ON and compared different stages of prostate cancer and metastatic tissue. They concluded that the methylation level increased in both CpG islands in parallel with more advanced cancer stages. They identified three CpG clusters with a high level of methylation, while there was no transcription. These results have been confirmed in the cell lines, too [61]. ERβ expression is lower in breast and ovarian tumors than in normal tissue. Therefore, it is assumed that ERβ has a tumor suppressor role [14,35,62,63,64,65]. DNA methylation regulates the expression of ERβ in breast and ovary cancers. Zhao et al. measured the CpG methylation level in the *ESR2* promoter region and the expression of ERβ in breast cancer cell lines and primary tumors. They showed that a lower level of ERβ expression is associated with ON promoter methylation, unlike the OK promoter, which was unmethylated in normal and tumor cells [66]. Rody et al. showed the increased level of methylation of the ON promoter. The treatment of cells with 5-AZAC was associated with an increased expression of ERβ [67]. In our study, we observed a significant association between ERβ promoter ON methylation and nodal metastasis in invasive breast cancer [68].

The other mechanism of epigenetic regulation of ERβ includes histone post-translational modifications. The important post-translational modifications of histones are acetylation, methylation, and phosphorylation. All of these modifications together result in transcription or an inhibition of transcription, depending on the aminoacid residue. Histone acetyltransferases (HATs) transfer acetyl groups to lysine residues in histones, leading to the uncoiling of chromatin and allowing for the accessibility of transcription factors and other regulators. Histone deacetylases (HDACs) remove the acetyl groups from histones. Histone methyltransferases (HMTs) add methyl groups to lysine or arginine residues in histones, while histone demethylases (HMDs) remove them [69]. There is not much information on the role of histone modifications in ERβ expression. Some studies showed that treatment with trichostatin, an HDAC inhibitor in breast, ovary, and prostate cancer cell lines, led to the re-expression of ERβ. The addition of 5′-aza-2-deoxycytidine (DNMT inhibitor) pronounced this effect even more, as reviewed in [70]. This means that histone deacetylases and DNA methylation supplement each other in inhibiting gene expression. Methylation can also act upon DNA-protein interactions and the activation of histone deacetylases, thus inducing chromatin condensation and the inactivation of gene expression [71,72]. In vitro studies show that DNA demethylation agents, especially combined with HDAC inhibitors, induce apoptosis, cell differentiation, and/or growth termination in lung, breast, prostate, and colon cells [73,74].

The small non-coding RNAs or microRNAs (miRNA) also participate in an important mechanism of epigenetic regulation of ERβ. Still, there is much to be learned about targeting ERβ by miRNAs. The most experimented with is miR-92, which targets the 3′-untranslated region of ERβ, thereby downregulating its expression. In MCF-7 cells, inhibiting miR-92 in a dose-dependent manner induces ERβ1expression. ERβ also regulates some miRNAs. It downregulates miR-145, miR-30a-5p, and miR-200a/b/429, which play a role in the inhibition of epithelial-mesenchymal transition. It upregulates miR-181a-5p, miR-10, and miR-375, which inhibit cholesterol biosynthesis in TNBC cells, regulate the composition of the extracellular matrix, and suppress proliferation, respectively. The studies so far show that miRNAs play an important role in regulating ERβ expression and cancer development (reviewed in [75]).

## 4. ERβ Role in the Metabolism of Cancer Cells

ERβ plays an important role in cancer cell metabolism. The studies on cell lines and animal models show that it participates in many metabolic pathways, as are lipid metabolism, oxidative phosphorylation, and glycolysis, as reviewed in [76]. In adipose tissue, ERβ, together with ERα, influences glucose and insulin metabolism, but estrogen predominantly binds to ERα [77]. Glycolysis is a very important process in tumor cells. Unlike normal cells, which metabolize glucose into lactate in anaerobic conditions, tumor cells do this in the presence of oxygen. Therefore, they, in some way, bypass the process of oxidative phosphorylation. In tumor cells, glucose intake increases, and the membrane glucose transporters (GLUTs) play a role in this process. Recent studies show that the key of glycolysis in tumors is the generation of the many intermediates they will use in the anabolic processes, which cancer cells need for their growth and energy sources [78]. In malignant mesothelioma cells, it was shown that ERβ represses succinate dehydrogenase B, a part of complex II of the electron transport chain of mitochondria. Therefore, ERβ lowers the activity of oxidative phosphorylation in mitochondria [79]. In ERα-negative breast cancer stem cells that expressed ERβ, adding its agonist, 2,3-bis(4-hydroxyphenyl)-propionitrile (DPN), increased glycolysis and lactate secretion in the growth media. In addition, oxygen consumption rates decreased, but this was reversed by adding ERβ antagonist 4-[2-phenyl-5,7-bis(trifluoromethyl)pyrazolo[1,5-a]pyrimidin-3-yl]phenol (PHTPP). In ERβ-negative cells, the expression of glycolytic genes decreased, but the expression of genes participating in oxidative phosphorylation increased. This indicates that ERβ in tumor cells accelerates glycolysis and inhibits oxidative phosphorylation. The authors of the study argue that ERβ is a possible target for therapy with antagonists [80]. However, Song et al. found that the upregulation of mitochondrial ERβ in TNBC cells activated oxidative phosphorylation [81]. 27-Hydroxycholesterol (27-HC) is a metabolite of cholesterol, and many studies have shown it is associated with BC. 27-HC is also a SERM that binds to ERα and ERβ, but with a higher affinity to ERβ, causing its conformational change. However, the observed association of plasma 27-HCT and ERβ in tumors is weak [82].

There is still much work to be done in research on the role of ERβ in the metabolic pathways of cancer cells and its possible utilization in therapy.

## 5. ERβ as a Potential Target in Clinical Practice

The important role of ERβ in carcinogenesis leads to the conclusion that it could be used in clinical practice as a potential therapeutic target. Several molecules that can bind to ERβ are being examined. One of them is genistein, a powerful phytoestrogen that is a natural constituent of a soy bean and binds to ERβ. It is a selective estrogen receptor modulator (SERM). Many in vitro and in vivo studies have shown the anticancer effects of genistein. However, many studies have also shown that genistein promotes cancer cell growth, as reviewed in [83]. Its potential to be used as an effective cancer treatment has been examined in a number of clinical studies. One of them investigated the effect of genistein on cytogenetic markers in postmenopausal women and showed a reduction of cytogenetic markers of cancer [84]. There was also a study where genistein combined with FOLFOX/FOLFOX-bevacizumab was proved safe for patients with colorectal cancer [85]. A clinical trial in phase 2, where genistein was used in prostate cancer patients, proved the safety and efficacy of genistein treatment [86]. The studies so far show that genistein could be a potential agent alone or in combination with other agents for the treatment of cancer. Several synthesized ERβ-selective agonists have been examined. Diarylproprionitryle, 8β-VE2, and SERBA-1 have all proved to prevent or reverse hyperplasia in prostate cancer animal models [87,88,89,90]. All these agonists could potentially be used in new therapies, but there is still much research to be conducted.

As we stated above, the main problem in hormone treatment is acquiring resistance to therapy in endocrine-related cancers. This has been seen in breast cancer (tamoxifen and aromatase inhibitors) and prostate cancer (anti-androgens). Other signaling pathways that bypass ER or AR signaling are probably being activated. One of the solutions to this problem is to target the cell growth PI3K⁄Akt⁄mTOR signaling pathway. This has been shown in advanced breast cancer, where inhibition of this pathway increased progression-free survival, as reviewed in [91].

## 6. ERβ in Endocrine Related Tumors

### 6.1. ERβ in Prostate Cancer and Its Metastases

The role of estrogen and estrogen receptors in the pathology of prostate cancer is well documented, especially that of ERβ. Epidemiological and experimental data indicate that dietary factors (related to estrogen exposition) are associated with prostate cancer incidence. Low consumption of animal fat and high consumption of phytoestrogen-rich food significantly decrease the risk for prostate cancer. As phytoestrogens are well-known ligands for ERβ, this is a probable explanation for the significantly lower incidence of prostate cancer in Asian countries [92,93]. The key to maintaining prostate tissue homeostasis is the testosterone/estrogen ratio. Any change of this ratio leads to a change in the prostate tissue in the tumor direction [94]. Preclinical and epidemiological studies show that the increased estrogen serum level or the increased estrogen/androgen ratio are associated with an increased risk of prostate cancer [95]. On the other hand, circulating testosterone level could be reduced by estrogen in prostate cancer patients and inhibit testosterone-induced advancement of cancer [96]. In normal and tumor tissue of the prostate, ERβ is abundant in stroma and epithelium, while ERα is expressed only in the stroma [50]. However, the role of estrogen receptors in prostate carcinogenesis remains questionable. Many studies have shown that ERβ acted as a tumor suppressor in prostate cancer cell lines and mouse models [90,97,98,99]. In ERβ-knockout mice, there was a high degree of proliferation and apoptosis in ventral prostate, suggesting that ERβ is a tumor suppressor. Activation of ERα influenced the proliferation and formation of premalignant lesions, while ERβ activation was antiproliferative [91]. On the other side, when a large cohort of prostate cancer samples was analyzed, ERβ correlated with decreased survival, suggesting ERβ was proliferative [100].

Fixemer and coworkers studied 132 prostate cancer patients and showed significantly decreased levels of ERβ expression in 30 of 47 patients with high-grade prostatic intraepithelial neoplasia [101]. In lymph node and bone metastases, ERβ was expressed at high and moderate levels. They concluded that untreated primary and metastatic prostatic adenocarcinoma retains ERβ expression, but in moderate and lower levels. On the contrary, hormone refractory tumors revealed decreased (38% of cases) or undetectable (13% of cases) rates of ERβ expression. In another study [102], Lai and coworkers analyzed 60 samples of bone and non-osseous metastases from 20 advanced androgen-independent prostate cancer patients. They also found retained ERβ expression in all metastases, but with variable intensities and percentages of ERβ-positive cells in non-osseous metastatic samples. Similar findings were reported by Leav and colleagues [53]. The presence of ERβ expression in prostate cancer, bone, and lymph node metastases suggests that ERβ could be a target for specific therapy, even in metastatic disease. The ERβ behavior in different studies is shown in Table 1.

Similar to breast cancer, epigenetic mechanisms play an important role in prostate carcinogenesis, too. While highly expressed in the normal prostate tissue, during pathogenesis of prostate cancer, ERβ expression gradually decreased [101,103]. Although the main mechanism of the loss of ERβ expression is not completely understood, some works showed that the hypermethylation of CpG-islands within the ERβ promoter ON might be involved in ERβ loss [61,74]. Zhu et al. showed that the methylation of *ESR2* gene promoter ON correlated with the loss of ERβ expression in primary prostate carcinomas, while ERβ was strongly expressed in metastasis. This study confirmed the previous finding that methylation could be reversed in metastatic tissues [53]. The changes in ERβ expression during prostate carcinogenesis imply that there is a switch that changes the role of ERβ from tumor-suppressive to proliferative [104].

The contradictory role of ERβ in prostate carcinogenesis could be, at least partially, attributed to the existence of ERβ isoforms, as are ERβ2 and ERβ5. ERβ2 and ERβ5 were linked with metastasis in samples from a prostate cancer patient cohort, as well as in the PC3 cell line [105]. In the prostate cancer cell line PC3, ERβ1 hindered proliferation and factors that act in bone metastasis, while ERβ5 acted the opposite [106]. However, ERβ1 seems to be lost in later stages of prostate cancer, while ERβ2 was expressed in those stages [91]. ERβ2 and ERβ5 isoforms can form heterodimers with ERβ1, consequently enhancing its transactivation [12]. Taking the complicated nature of ERβ isoforms into account, we need more research to elucidate their role in prostate carcinogenesis.

**Table 1 ijms-22-01656-t001:** ERβ function (tumor suppressive or proliferative) and observed associations in studies in different cancer types: breast, prostate, thyroid, and colon.

Cancer Type	ERβ Function			ERβ Associations in Studies	
Breast	ERα+	Tumor-suppressive	Omoto et al. [91,107] Secreto et al. [108] Haldosen et al. [109]	No correlation with clinical variables	Hopp et al. [110] De Cremoux et al. [111] Stefanou et al. [112] Madeira et al. [113]
	Erα−	Tumor-suppressive	Gruvberger-Saal et al. [114] Honma et al. [115,116] Reese et al. [117] Schuler-Toprak et al. [118]	There is a correlation with clinical variables	Reviewed in Zhou et al. [75]
		Proliferative	Austin et al. [119] Piperigkou et al. [120]		
Prostate	Tumor-suppressive		Cheng et al. [97] Dey et al. [90] Hussain et al. [98] Wu et al. [99]	ERβ↑	Decreased BFFS: Grindstad et al. [100]
	Proliferative		Grindstad et al. [100]	ERβ↓	High grade neoplasia: Fiexemer et al. [101] Metastasis:Lai et al. [102] Leav et al. [53]
Thyroid	Tumor-suppressive		Dong et al. [121] Zeng et al. [122] Cho et al. [123]	ERβ-negative → Poor outcome: Heikkilä et al. [124] Recurrence: Ahn et al. [125] Nodal metastasis: Dong et al. [126]	
Colon	Tumor-suppressive		Hartman et al. [127] Giroux et al. [128] Saleiro et al. [129]	ERβ↓	Tumor progression: Elbanna et al. [130] Foley et al. [131] Konstantinopoulos [132] Jassam et al. [133]
				ERβ-	Poorer OS and DFS: Rudolph et al. [134] Fang et al. [135]

ERβ↑: ERβ increasing; ERβ↓: ERβ decreasing; BFFS: Biochemical failure free survival; OS: overall survival; DFS: disease free survival.

### 6.2. ERβ in Breast Cancer and Its Metastases

Until today, the immunohistochemical status of estrogen receptor alpha (ERα), progesterone receptor (PR), and human epidermal growth factor receptor-2 (HER2) has been widely used as a diagnostic tool, based on which the therapy for breast cancer is determined. Statuses of these receptors define four main subgroups of breast cancer (BC): Luminal A (ER- and PR-positive and HER2-negative), Luminal B (ER-, PR-, and HER2-positive), HER2 overexpressed (ER- and PR-negative and HER2-positive), and triple-negative breast cancer (TNBC) (ER-, PR-, and HER2-negative) [109]. Despite the obvious progress in the research of the molecular signature of different BC subgroups and histological types, tamoxifen is still the first line of therapy for ERα-positive patients with a 70% response. One of the major problems in treating breast cancer is the acquiring resistance to standard therapies. Clinicians and researchers mainly agree that there is a need for the introduction of additional markers in diagnosis and therapy. Another important issue is TNBC, which is defined by the lack of expression of estrogen receptor alpha, progesterone receptor, and human epidermal growth factor receptor-2. Many studies have reported the expression of ERβ in TNBC (ERα-negative/ERβ-positive status), as reviewed in a paper by Chen and Russo [136]. The majority of data from the research on clinical samples and cell lines suggest that ERβ has antiproliferative, pro-apoptotic, and tumor-suppressive functions [35,137,138,139]. In ERα-positive cases, ERβ mainly acts as a tumor-suppressor [91,109]. On the other hand, in ERα-negative breast cancer, findings are controversial: some studies found ERβ was tumor-suppressive [114,116,117,118], while others found that ERβ is proliferative [119,120]. Wisinski et al. assumed that ERα-negative patients could benefit from therapy with estrogen, as it has already been used in ERα-positive cases resistant to standard therapy. However, their clinical study from 2016, which examined the efficacy of oral estrogen in TNBC patients, was closed, because of the disease progression in most of the patients. The authors of the study assumed that the failure of estrogen therapy was due to the fact that there was no selection based on the ERβ status among patients. [140]. Consequently, Mayo Clinic researchers opened a phase II clinical trial to test the effect of estradiol only in ERβ-positive metastatic TNBC women (ClinicalTrials.gov Identifier: NCT03941730) [117,141]. In tissue microarrays of breast cancer, ERβ expression was significantly associated with ERα and PR and was higher in luminal A and luminal B subtypes compared to HER2+ and basal-like subtypes. ERβ inversely associated with HER2 and EGFR expression [142]. This proves that ERβ could be a potential target in BC therapy in ERα-positive tumors. Zhou and al. reviewed the latest studies showing the association of clinical outcomes in breast cancer patients and the status of ERα and ERβ [75]. In TNBC, ERβ expression was significantly lower than in the triple-positive breast cancers [143]. The results of the studies in the TNBC group are inconsistent. Some studies showed no association between ERβ1 expression and OS and DFS in the TNBC group of patients [144]. Some studies showed that ERβ1 was associated with better OS, DFS, and distant metastasis-free survival [145], while others have reported the contrary [119,143,146]. The ERβ role and associations in the different types of breast cancer in various studies are shown in Table 1. Unfortunately, the clinical value of ERβ regarding prediction, prognosis, and its use as a possible therapeutic target is not established yet. Due to the methodological limitations caused by the unspecific commercial antibodies, the complex biological organization of the *ESR2* gene, and its expression, most research on clinical samples still produces controversial data. The differences in the starting material for the study are also amongst the causes of controversy. Measuring ERβ mRNA by qPCR allows for a precise quantification of each isoform and internal exon-deleted splice variant. Results from such studies are not concordant with those performed on tumor samples by using antibodies. Besides the use of nonspecific antibodies, posttranslational modifications of ERβ protein are also amongst the reasons for this.

Elucidating the role of ERβ requires an understanding of its complexity, such as the existence of ERβ genetic variants and splice variants. ERβ genetic polymorphisms have a questionable role in breast cancer. Commonly studied ERβ polymorphisms, rs4986938 and rs928554, in the 3’-untranslated region of *ESR2*, were not associated with any changes on mRNA or protein level [147]. However, in one meta-study, the rs4986938 polymorphism was associated with decreased breast cancer risk [148]. The role of ERβ genetic variants in breast cancer should be further examined.

There are five isoforms of ERβ identified so far that result from alternative splicing of the two last coding exons of *ESR2* [5,11,12,109,149]. The most examined isoform, ERβ2, was shown to inhibit transactivation of ERα and was significantly correlated with the ERα-negative phenotype [150]. It is possible that, by forming heterodimers with ERα, ERβ2 induces its degradation [109]. However, the role of ERβ2 in therapy response is not yet confirmed. Nuclear ERβ2 was shown to have a good predictive value of therapy response [151]. ERβ2 mRNA was associated with better response to tamoxifen therapy in ERα-positive patients [152]. On the other hand, in the small number of primary lesion samples, ERβ2 was correlated with poor tamoxifen response in cases with low PR expression [153].

Understanding the role of ERβ is complicated further by the epigenetic mechanisms, as are the hypermethylation of *ESR2* gene promoter and histone acetylation. The *ESR2* gene alternatively transcribes from two promoters, ON or OK, that results in two ERβ mRNA isoforms [61,66]. Hypermethylation of the ON promoter was shown to be the cause of *ESR2* gene silencing in breast cancer [66,67]. It seems that *ESR2* gene silencing plays a role in cancer progression. Comparing breast cancer tissues and adjacent normal tissues, Gao et al. found a significant difference in methylation level and mRNA expression. There was also a significantly lower expression of mRNA in pre-cancerous tissue compared to tissue in a more progressive breast cancer state [154]. Some researchers and our group came to the conclusion that the hypermethylation of the *ESR2* gene ON promoter was associated with more progressed disease [67,68]. Based on the previous studies, hypermethylation could be a potential additional diagnostic/therapeutic marker that could be targeted by demethylation agents.

Expression of ERβ is probably regulated by histone acetylation, too. Enzymes, i.e., histone acetyltransferases, transfer an acetyl group to the histones, thereby lowering affinity of histones for DNA. The tightly coiled DNA untangles and becomes transcriptionally active [155]. If the ERβ is a tumor suppressor, patients will surely benefit from the therapy, which would restore ERβ expression. Accordingly, histone deacetylase inhibitors (HDI) increased the expression of ERβ and decreased the expression of ERα in breast cancer cell lines [156]. In one study, HDI-trichostatin induced the expression and nuclear translocation of ERβ in TNBC cells, leading to increased tamoxifen response [157]. In a clinical trial, combination therapy of vorinostat and tamoxifen in patients with advanced breast cancer appeared promising because 40% of patients manifested disease stabilization [158].

The role of microRNAs (miRNAs) in cancer is still largely unexplored. miRNA-92 expression significantly correlated with the ERβ1 expression in samples of invasive breast cancer and in cell lines. It is assumed that miRNA-92 inhibits ERβ1 expression by binding to the 3’-untranslated region of ERβ1-mRNA [159]. Besides miRNA-92, there are probably other miRNAs that regulate the expression of ERβ1 and act in the process of breast carcinogenesis.

The ERβ role is complicated even more by the interaction with ERα. ERβ/ERα heterodimers have different gene targets from the homodimers of these receptors, as discussed in [160]. In addition, the expression level of these receptors determines the direction of different signaling pathways [75]. The increased level of ERβ decreases the level of ERα by downregulating its transactivation [161]. Considering that ERα and ERβ both mediate estrogen effects, there is an emergent need to define a new receptor status for breast cancer in clinical routines that include the status of ERβ1 and/or ERβ2 [162]. Despite the established clinical management of ERα-positive breast cancer, the inclusion of the new receptor ERβ will probably significantly improve adjuvant hormonal treatment. It has become clear that the main change during the progression of breast cancer disease is a declining ERβ/ERα ratio, from normal epithelium to ductal and lobular carcinomas. In breast cancer, the presence of lymph node metastasis with a large tumor size is associated with poor prognosis. Initially, a metastatic cell retains its ERα positivity. If a primary tumor is ERα-positive, more than 80% of the lymph node metastasis retains ERα [163], although gene expression profiling and immunohistochemical analyses have shown the differential regulation of genes by estradiol in tumors and matched lymph node metastasis [164]. Similar statuses of ERα, ERβ1, and ERβ2 in primary tumors and their paired sentinel lymph nodes have been reported by Rosin et al. [162]. However, they found a more frequent expression of ERβ2 among node-positive tumors, and that ERβ2 expression in metastasis is associated with poor survival.

The benefits of combinational therapies, e.g., standard therapy with demethylation agents and histone deacetylase inhibitors, should be evaluated in future clinical studies.

### 6.3. Estrogen Receptors in Thyroid Cancer

Thyroid cancer is the most common endocrine malignancy [165]. A significant gender disparity has been observed in the incidence, aggressiveness, and prognosis of this type of cancer. There is a female predominance of around three times in incidence of mainly less aggressive histologic subtypes of thyroid cancer, while the more aggressive subtypes have similar gender distribution. Female predominance is seen only in women of reproductive age, while postmenopausal women have the same incidence rates as men [166]. The molecular basis of this gender disparity is still unclear, but a number of studies point to the role of estrogens and their receptors in the pathogenesis and progression of thyroid cancer. Estrogen was found to act as a potent growth stimulator of benign and malignant thyroid cells in in vitro studies [122,167,168,169,170].

Estrogen receptors ERα and ERβ have been detected in both normal and neoplastic thyroid tissues. While some studies detected only one of the isoforms [171,172,173], most studies show that thyroid cancer cells express both estrogen receptor isoforms [123,169,174,175]. Both receptor isoforms were also detected in human thyroid stem/progenitor cells by immunofluorescence and quantitative PCR, with ERα mainly localized in the nucleus and ERβ mainly in the perinuclear region of the cytoplasm [176].

The significance of different patterns of distribution and expression of ERα and ERβ in thyroid carcinoma has been proposed [177,178,179]. As postulated in breast cancer, estrogen binding to ERα would promote cell proliferation and growth, and through ERβ binding, it would induce apoptosis and promote differentiation and other suppressive functions in thyroid tumors.

In addition to a well-documented growth promoting effect of estrogens on human thyroid cells, another indicator of the involvement of estrogens in thyroid cancer pathogenesis is the finding that the level of expression of ERs in tumor compared to benign or normal thyroid tissue is frequently altered [172,174,175,179,180,181,182,183,184]. A contributing role of the imbalance between ERα and ERβ in thyroid carcinogenesis was also suggested [123,174,175].

While some studies have not found significant correlations between ER expression/reactivity and clinical and pathological parameters in thyroid cancer patients [182,185], a number of studies have. In follicular thyroid cancer, ERα expression was associated with well differentiated tumors and reduced the incidence of disease recurrence [186]. In papillary thyroid cancer patients, ERα expression was positively correlated with tumor size and a more favorable outcome [184,187]. A study of ERα expression in primary and metastatic papillary thyroid carcinoma found significantly higher positivity in metastatic tumors [188]. In addition, a higher ERα expression and a higher ERα/ERβ ratio were associated with more aggressive features and worse overall survival in female PTC patients [179]. A loss of ERβ expression was associated with poor outcomes in patients with follicular thyroid carcinoma [124] and with recurrence in young female PTC patients [125]. According to a study by Magri et al., ERβ negativity was associated with a presence of vascular invasion in small differentiated thyroid carcinoma [183]. Moreover, reduced expression of ERβ was associated with nodal metastasis and extra-thyroid spread in female ERα-negative papillary thyroid carcinoma patients [126]. A lower expression level of ERβ was also described in undifferentiated thyroid stem/progenitor cells compared to differentiated thyroid cells [176]. The ERβ role in these studies is presented in Table 1.

Few studies have investigated the expression of ERβ variants in human thyroid tissue. Egawa et al. have investigated the expression of several ERβ mRNA (β1, β2, and β5) in normal thyroid and various neoplastic thyroid tissues and found no significant differences in the proportions of ERβ mRNA variants [174]. On the other hand, the results of Dong et al. show that the expression patterns of ERβ1 and ERβ2 differ between papillary thyroid cancer and benign thyroid tissue, and the expression level of ERβ2 was found to be lower in female patients of reproductive age with lymph node metastasis than in patients without lymph node metastasis [189,190].

Although there is strong evidence that estrogen is involved in the pathogenesis of thyroid cancer in females, the results are often contradictory, showing widely varying patterns of ER expression as well as a loss or overexpression of certain receptor isoforms. Taking into account the high complexity of ER expression, the existence of multiple variants, epigenetic modifications, interactions with microRNAs, different subcellular localizations, and multiple phosphorylation sites, it is possible that a comparison of ER expression patterns alone is not enough to draw conclusions regarding the relevance of estrogen and ERs in thyroid cancer pathogenesis. The pathways involved are complex and poorly understood, and merit further investigation.

## 7. ERβ in Non-Endocrine Tumors

### ERβ in Colorectal Cancer

ERβ is the predominant estrogen receptor expressed in both normal and malignant colorectal mucosa [130,191]. However, during colorectal carcinogenesis, ERβ expression declines, showing an inverse relationship with tumor progression [4,131,132]. Numerous studies conducted on cell lines and clinical samples showed that the reduced expression of ERβ is related to an advanced tumor stage and grade of colorectal cancer (CRC) [127,130,133,134]. Associations with other characteristics of poor prognosis, including loss of differentiation, vascular invasion, and decreased apoptosis, were also observed [130,132,133]. Studies that examined the prognostic implication of ERβ expression in CRC revealed that patients with ERβ-negative tumors had significantly poorer OS compared to ERβ-positive tumors [134,135]. In addition, the data obtained by Rudolph and coworkers demonstrated that the absence of ERβ expression was associated with an increased risk of CRC-specific death and a worse disease-free survival (DFS) [134]. Obtained data indicate that estrogen-mediated signaling through ERβ exerts a protective role against colorectal carcinogenesis. This could be important for designing new therapeutic approaches targeting the estrogen pathway in CRC, especially in ERβ-positive tumors [192]. Studies showing the predominantly tumor-suppressive role of ERβ in colon are shown in Table 1.

The protective role of ERβ signaling in the development of CRC has been studied and established in animal models. A number of in vitro studies in mice demonstrated that the loss of ERβ leads to hyper-proliferation, decreased apoptosis, inflammation, and dedifferentiation [128,129,193]. Accumulated experimental data have demonstrated that the tumor-suppressive action of ERβ could be exerted through genomic and non-genomic effects of estrogen on target genes in epithelial colonocytes [194]. Genomic effects involve the activation of gene transcription through either direct interaction with specific DNA sequences in the promoter of target genes, known as estrogen response elements (EREs), or interaction with other transcription factors such as SP1, AP1, and NFκB [195,196,197,198,199]. The non-genomic effects of estrogen are exerted through modulation of particular cellular processes through the activation of certain signaling pathways without direct interaction with DNA [200,201,202,203,204,205,206]. Nevertheless, observed data suggest that the loss of ERβ alone might not be sufficient to promote colorectal carcinogenesis; rather, it accelerates tumorigenesis by eliminating a protective mechanism against genotoxic stress in the early phases of epithelial colonic cell transformation [192].

A full understanding of estrogen-mediated signaling involved in tumor suppression is further complicated by the presence of five alternative isoforms of ERβ. Existing experimental and clinical data indicate that ERβ2 and ERβ5 are the predominant isoforms identified in CRC samples [207], but their specific role in CRC is not fully elucidated.

Important results obtained in human colon cancer cell lines indicated that ERβ exerts antitumorigenic effects in several ways. Some of these mechanisms include the reduction of cell proliferation through the modulation of cell cycle regulators [127,208], the induction of apoptosis through increased p53 signaling [209], and the upregulation of DNA mismatch repair genes [210]. A recent study on cell lines and animal models demonstrated that ERβ can repress the metastatic potential of CRC cells through the upregulation of miRNA-205 and the subsequent downregulation of *POX1* oncogene expression [211].

The precise mechanisms of the loss of ERβ expression in CRC are not fully established. Foley and coworkers found no difference in ERβ mRNA expression between normal and malignant colorectal mucosa samples, which supports a posttranscriptional mechanism for ERβ protein downregulation [131]. Issa and coworkers found that the ER alpha gene is methylated in 90% of colon cancer tissues. Passarelli and coworkers found that single-nucleotide polymorphisms (SNPs) located in the 5′-regulatory region of the ERβ gene could be markers of receptor isoform expression or CpG island methylation and transcriptional inactivation of ERβ [212]. These authors suggest that SNPs in the ERβ promoter could be related to tumor progression and metastasis. In addition, there is a possibility that changes in the expression levels of specific transcriptional regulators could be involved in the modulation of ERβ transcription in CRC [213]. Data obtained from the study of Honma demonstrated that ERβ gene cytosine-adenine (*ESR2* CA) repeat polymorphism could be related to CRC risk through impairing ERβ signaling and lower wild-type ERβ expression [214].

On the contrary to the BC, there is ample epidemiological evidence that confirms the protective role of estrogen signaling in the development of CRC, as reviewed in [215]. Colorectal cancer incidence and mortality rates are lower in females than in males, and numerous epidemiological studies suggest that hormone replacement therapy (HRT) reduces cancer risk [216,217,218,219,220,221] and the risk of cancer-related death [222,223,224,225] in postmenopausal women. The existing preclinical data established that the protective effect of estrogen in colon epithelial cells is exerted through ERβ, as reviewed in [226]. Moreover, there is much evidence suggesting that ERβ signaling might be responsible for the observed gender-specific differences related to proximal versus distal CRC and the associated differences in genetic changes. It has been shown that the reduction of ERβ expression is more prominent in proximal than in distal CRC cases. It is well known that women develop more proximal colon and rectal tumors, while men develop more distal ones [133]. The results from the recent study of Missiaglia and coworkers showed an association of the microsatellite instability (MSI) phenotype with a proximal location, a methylator (CIMP) phenotype, and an inflammatory subtype of CRC [227]. It has been previously shown that endocrine factors are associated with MSI tumors and that women are more likely than men to have MSI tumors at an older age [228]. These findings were addressed to the ability of estrogen to upregulate DNA mismatch repair genes via ERβ and thus preserve genome stability in colonic epithelia cells [210,228,229]. In addition, data collected from a number of additional epidemiological studies on post-menopausal women suggest that estrogen could reduce the risk of specific subtypes of CRC defined by the expression of cell cycle regulators, DNA methylation, and somatic mutations of particular genes as well as MSI status [230,231,232,233].

Similar to prostate cancer, the incidence of CRC is lower in Asia, which should be ascribed to the extensive use of soy products containing ERβ ligands, i.e., phytoestrogens. In the meta-analysis conducted by Yan and coworkers, soy consumption was found to be associated with a decreased risk of CRC in women but not in men, which suggest that phytoestrogens are more effective when combined with endogenous estrogens [234,235]. Nevertheless, the results from two independent studies [213,216] suggest a controversial role for estrogen in CRC. These groups of authors demonstrated that estrogen may have a protective effect initially, but in later stages it can stimulate proliferation. This implicates a complex role for estrogen in CRC, which requires additional investigations and should be taken into account when considering new therapeutic approaches. Armstrong and coworkers, in their study on mice, found that this controversial role of estrogen could be ascribed to the observed upregulation of ERα, concurrent with reduced ERβ expression in later stages of colorectal carcinogenesis [236]. This could be a possible explanation for altered response to estrogen exposure due to the dietary factors and/or hormone replacement therapy, as described in the previously mentioned studies.

Elucidating the mechanisms of ERβ antitumorigenic activity has significant potential for the clinical management of CRC patients and the prevention of disease in those at risk. The activation or upregulation of ERβ in colonocytes by phytoestrogens and synthetic ERβ selective agonists are possible therapeutic approaches.

## 8. Conclusions

Currently, one cannot claim that ERβ is always tumor-suppressive and that ERα is proliferative. Many factors influence how ERβ behaves in a cell, in terms of cell type, stage of carcinogenesis, the other receptors and regulators, the existence of ERβ isoforms, and epigenetic mechanisms. This complex mechanism cannot be observed separately from the organism. More research is needed to investigate the role of ERβ in normal physiology, as well as in carcinogenesis. The existence of different expression levels for each isoform as well as internal exon splice variants are amongst the most important future challenges in the biology and pathology of ERs.

We assume that the status of ERβ could be clearer. First, a standardization of research techniques is needed to compare different studies. Second, highly selective ERβ antibodies or methods should be developed and evaluated in a large number of well-characterized cancer samples. In the case of ERs, this is particularly important for the quantification of internal exon deleted variants. Third, future studies should pay more attention to specific ERβ isoforms and to epigenetic mechanisms shown to play a role in carcinogenesis.

The assessment of biomarkers is incomplete without taking into account microenvironment of every patient. Therefore, the goal of the diagnostics and treatment of cancer should be personalization. One will only accomplish that by analyzing the genetic and protein landscape of every patient, leading to decision-making about therapy accordingly.

## Figures and Tables

**Figure 1 ijms-22-01656-f001:**
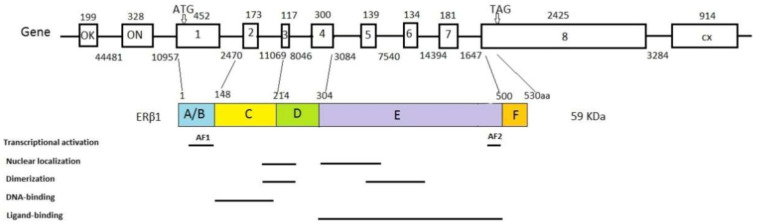
Estrogen Receptor Beta (Erβ) gene, protein structure, and functional domains. Gene: The boxes represent exons; the lines are intrones; the numbers above the boxes represent the exon sizes (bp); the numbers below the lines indicate the introne sizes (bp); the lines between the gene and protein indicate protein domain junctions. ERβ1 protein: The numbers above the boxes represent the size of the protein domains (the number of amino acids). The figure is modified under CC BY NC, based on [17].

**Figure 2 ijms-22-01656-f002:**
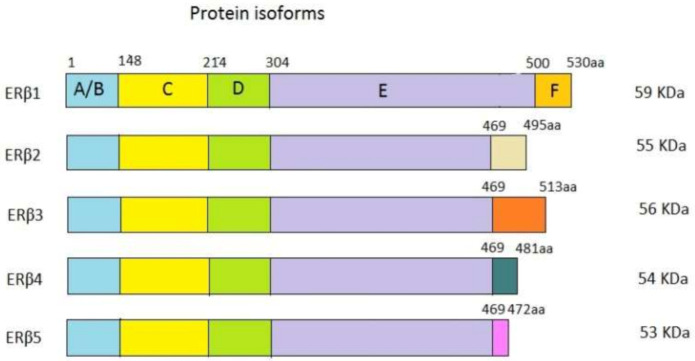
ERβ isoforms—the wt protein isoform ERβ1 and the other isoforms from ERβ2 to ERβ5. The boxes indicate protein domains. The numbers above the boxes represent the sizes of domains (the number of amino acids). The numbers on the right side of the isoforms indicate the molecular weight of every isoform in kDa. The figure is modified under CC BY NC, based on [17].

## Data Availability

No new data were created or analyzed in this study. Data sharing is not applicable to this article.

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
