# Peer review of "Estrogen Receptor Beta: The Promising Biomarker and Potential Target in Metastases"

_ijms, 2021, doi:10.3390/ijms22041656_

Round 1
Reviewer 1 Report
This review article by Božović et al., discusses the heterogeneity in the expression and regulation of Estrogen receptor beta in endocrine and non-endocrine tumors. The authors has efficiently summarized experimental outcomes available in this areas however, the following observations can be used for further augmentation of this review article.
- Authors should explain the structural details of different isoforms of Estrogen receptor beta through an illustration.
- It is imperative to discuss the molecular pathways of Estrogen receptor alpha and beta in cancer, and highlight the divergent role in pro and anti-apoptotic pathways.
- Very little information included regarding epigenetic regulation of Estrogen receptor beta, authors should separately discuss the role of epigenetic factors in the regulation of its expression.
- Include a precise section describing the significance of Estrogen receptor beta signaling in cancer cells metabolism.
- As the titled highlights this molecule as a potential therapeutic target, it is better to include some information regarding the clinical implications of targeting.
Author Response
The response to Reviewer 1
Remark 1: “Authors should explain the structural details of different isoforms of Estrogen receptor beta through an illustration.”
Response: We agree with the reviewer, thus we added the information on the ERβ isoforms, in the Section 1 of the Introduction (lines 35-74). We also added the Figure 1, titled “ERβ gene, protein structure, and functional domains” and the Figure 2, titled “ERβ isoforms—the wt protein isoform ERβ1 and the other isoforms from ERβ2 to ERβ5”, in Section 1.
Remark 2: “It is imperative to discuss the molecular pathways of Estrogen receptor alpha and beta in cancer, and highlight the divergent role in pro and anti-apoptotic pathways.”
Response: We fully agree with the reviewer´s comment on the molecular pathways. In the revised manuscript, we included an additional section 2, titled “The molecular pathways of Estrogen receptor alpha and beta in cancer”, lines 89-130.
Remark 3: “Very little information included regarding epigenetic regulation of Estrogen receptor beta, authors should separately discuss the role of epigenetic factors in the regulation of its expression.”
Response: The reviewer makes an excellent point on the epigenetic factors in the regulation of ERβ expression. According to the reviewer´s suggestion we included a separate section 3, titled “Epigenetic regulation of ERβ” (lines 132-233) where we thoroughly discuss different mechanisms of epigenetic regulation of ERβ in cancerogenesis. We also cited our paper about the methylation of ERβ ON promoter in invasive breast cancer.
Remark 4: “Include a precise section describing the significance of Estrogen receptor beta signaling in cancer cells metabolism.”
Response: The reviewer is right. Thus, we added Section 4, titled “ERβ role in the metabolism of cancer cells”, lines 235-263.
Remark 5: “As the titled highlights this molecule as a potential therapeutic target, it is better to include some information regarding the clinical implications of targeting.”
Response: According to a reviewer’s remark, we added Section 5, titled “ERβ as a potential target in clinical practice”, lines 265-288, in which we discuss the possibility of utilization of ERβ in cancer therapy.
Reviewer 2 Report
The review is poorly organised and written. Would benefit from the inclusion of figures to show structure and information on the different ERb isoforms in the introduction. Some information is included in lines 137-144 but this can expanded. Some summary tables for the different cancers would greatly improve the review. The sections on line 38-46 is not really about ERb in prostate cancer. Similarly lines 96-106 are not relevant to ERb in breast cancer. What is known about ERb in the different breast cancer subtypes?
Author Response
The response to Reviewer 2
Remark: “Would benefit from the inclusion of figures to show structure and information on the different ERb isoforms in the introduction. Some information is included in lines 137-144 but this can expanded”
Response: We thank the reviewer for this comment. This comment of the Reviewer 2 is in concordance with the first remark of Reviewer 1, and accordingly we added the corresponding information in Section 1 of the Introduction (lines 35-74). We also added the Figure 1, titled “ERβ gene, protein structure, and functional domains” and the Figure 2, titled “ERβ isoforms—the wt protein isoform ERβ1 and the other isoforms from ERβ2 to ERβ5”, in Section 1.
Remark: “Some summary tables for the different cancers would greatly improve the review.”
Response: According to the reviewer´s suggestion we included summary table in line 344, titled “Table 1. ERβ function (tumor suppressive or proliferative) and observed associations in studies in different cancer types: breast, prostate, thyroid, and colon.”, and indicated it in the text in the lines 323,386, 501, and 536.
Remark: “The sections on line 38-46 is not really about ERb in prostate cancer. Similarly lines 96-106 are not relevant to ERb in breast cancer”
Response: We agree with reviewer’s comment, thus we deleted the lines 38-41. However, in the lines 96-106 (now 350-360) we introduce the significance of classical receptor status (ERα, PR, HER2) in clinical management of BC patients, and this few sentences are followed with the text which point out the possibility that ERβ might be a potential therapeutic target in “triple negative/ERβ positive” breast cancer. We think that this introduction is relevant to the rest of the text about ERβ role in breast cancer.
Remark: “What is known about ERb in the different breast cancer subtypes?”
Response: We thank the reviewer for this comment. We have already discussed the ERβ role in ERα- positive, and ERα-negative breast cancer types, in the lines 106-121 (now 360-375). However, we added the paragraph from line 375-386, and the Table 1, in line 344, in which we summarize the information about the role and associations of ERβ in different subtypes of BC.
Round 2
Reviewer 1 Report
Authors have made all the changes in the manuscript as suggested, the revised version is improved.
Reviewer 2 Report
The figures and table have significantly improved the manuscript